# Humoral Responses Elicited by SARS-CoV-2 mRNA Vaccine in People Living with HIV

**DOI:** 10.3390/v15102004

**Published:** 2023-09-26

**Authors:** Lorie Marchitto, Debashree Chatterjee, Shilei Ding, Gabrielle Gendron-Lepage, Alexandra Tauzin, Marianne Boutin, Mehdi Benlarbi, Halima Medjahed, Mohamed Sylla, Hélène Lanctôt, Madeleine Durand, Andrés Finzi, Cécile Tremblay

**Affiliations:** 1Centre de Recherche du CHUM, Montreal, QC H2X 0A9, Canada; 2Département de Microbiologie, Infectiologie et Immunologie, Université de Montréal, Montreal, QC H3C 3J7, Canada

**Keywords:** SARS-CoV-2, PLWH, humoral responses, antibody responses, ADCC, neutralization, variants of concern

## Abstract

While mRNA SARS-CoV-2 vaccination elicits strong humoral responses in the general population, humoral responses in people living with HIV (PLWH) remain to be clarified. Here, we conducted a longitudinal study of vaccine immunogenicity elicited after two and three doses of mRNA SARS-CoV-2 vaccine in PLWH stratified by their CD4 count. We measured the capacity of the antibodies elicited by vaccination to bind the Spike glycoprotein of different variants of concern (VOCs). We also evaluated the Fc-mediated effector functions of these antibodies by measuring their ability to eliminate CEM.NKr cells stably expressing SARS-CoV-2 Spikes. Finally, we measured the relative capacity of the antibodies to neutralize authentic SARS-CoV-2 virus after the third dose of mRNA vaccine. We found that after two doses of SARS-CoV-2 mRNA vaccine, PLWH with a CD4 count < 250/mm^3^ had lower levels of anti-RBD IgG antibodies compared to PLWH with a CD4 count > 250/mm^3^ (*p* < 0.05). A third dose increased these levels and importantly, no major differences were observed in their capacity to mediate Fc-effector functions and neutralize authentic SARS-CoV-2. Overall, our work demonstrates the importance of mRNA vaccine boosting in immuno-compromised individuals presenting low levels of CD4.

## 1. Introduction

The emergence of the severe acute respiratory syndrome coronavirus 2 (SARS-CoV-2) in late 2019 led to the development of preventive and curative strategies worldwide. Among them, messenger RNA (mRNA) vaccines have shown great efficiency as a preventive approach [1,2,3,4]. mRNA SARS-CoV-2 vaccines were first manufactured to elicit antibodies (Abs) against the Wuhan ancestral Spike (S). More recently, the development of bivalent vaccines, encoding for both the Wuhan ancestral Spike and the Omicron subvariant Spike (BA.1 or BA.4/5), aim to increase the vaccine’s efficacy against several variants of concern (VOCs) [5,6] and were shown to be well tolerated in the global population [7,8].

Despite the proof-reading activity of the nonstructural protein nsp14 protein, the SARS-CoV-2 virus harbors between 10^−6^ and 10^−4^ substitutions per nucleotide per cell infection [9,10], promoting the rise of VOCs. Therefore, D614G was the first variant that emerged from the Wuhan ancestral Spike and spread worldwide [11]. The continuous emergence of different VOCs could be exemplified by the spread of Alpha, Beta, Gamma, Delta, Omicron and Omicron subvariants which presented increased resistance to the neutralization of vaccine-elicited antibodies [12,13,14,15,16].

Albeit SARS-CoV-2 vaccines are efficient in the global population, the humoral responses and vaccine immunogenicity in immunocompromised individuals were shown to be affected [17,18,19,20,21]. It was previously shown that antibody responses and protective immunity are weakened in transplant and hemodialysis patients, where the quantitative and qualitative humoral responses are poorer than the low-risk individuals [17,18]. These differences highlight the heterogenicity of humoral responses after vaccination among individuals. However, despite recent studies on the topic, less is known about the humoral responses developed after SARS-CoV-2 vaccination in people living with HIV (PLWH) [22,23].

We previously described the humoral responses in PLWH stratified by their CD4 count after one dose of mRNA vaccine and demonstrated the reduced anti-receptor binding domain (RBD) IgG responses in individuals with a CD4 count below 250/mm^3^, suggesting the requirement of a vaccination boost to improve humoral responses in these individuals [24]. We were interested in evaluating whether a vaccination boost had a differential effect according to CD4 count. Here, we conducted a longitudinal study using the same cohort of PLWH stratified by their CD4 count and assessed the quantity and quality of anti-SARS-CoV-2 antibodies elicited after the second and third dose of vaccination.

## 2. Materials and Methods

### 2.1. Ethics Statement

All work was conducted in accordance with the Declaration of Helsinki in terms of informed consent and approval by an appropriate institutional board. Blood samples were obtained from donors who consented to participate in this research project at the Centre de Recherche du CHUM and approved by the CHUM Research Ethics Board (#MP-02-2021-9513, 20.424). Plasma were isolated by centrifugation and Ficoll gradient and samples were stored at −80 °C until use.

### 2.2. Human Subjects

The study was conducted in 40 PLWH (33 males and 7 females; age range: 25–77 years) vaccinated with three doses of SARS-CoV-2 vaccine; 7 PLWH (5 males and 2 females; age range: 25–77 years) had a CD4 count < 250/mm^3^; 16 individuals (14 males and 2 females; age range: 40–62 years) had a CD4 count between 250 and 500/mm^3^ and 17 individuals (14 males and 3 females; age range: 35–60 years) had a CD4 count > 500/mm^3^. All this information is summarized in Table 1. No specific criteria such as number of patients (sample size), gender, sex, clinical or demographic were used for inclusion.

### 2.3. Plasma and Antibodies

Plasma from PLWH were collected, heat-inactivated for 1 h at 56 °C and stored at −80 °C until ready to use in subsequent experiments. Plasma from uninfected donors collected before the pandemic were used as negative controls and used to calculate the seropositivity threshold in our enzyme-linked immunosorbent assay (ELISA), antibody-dependent cellular cytotoxicity (ADCC) and flow cytometry assays (see below). The RBD-specific monoclonal antibody CR3022 was used as a positive control in the ELISA assay and was previously described [15,16,25,26,27,28,29]. Horseradish peroxidase (HRP)-conjugated specific for the Fc region of human IgG (Invitrogen, Thermo Fisher Scientific, Waltham, MA, USA) was used as a secondary Abs to detect Ab binding in ELISA. The conformationally independent S2-specific mAb CV3-25 was used as a positive control in flow cytometry assays. Alexa Fluor-647-conjugated goat anti-human Abs able to detect IgG isotypes (anti-human IgG Jackson ImmunoResearch Laboratories, West Grove, PA, USA) was used as a secondary Ab to detect plasma binding in flow cytometry experiments.

### 2.4. Cell Lines

293T human embryonic kidney and Vero E6 transmembrane serine protease 2 (TMPRSS2) [30,31] were maintained at 37 °C under 5% CO_2_ in Dulbecco’s modified Eagle’s medium (DMEM) (Wisent) containing 5% fetal bovine serum (FBS) (VWR) and 100 μg/mL of penicillin-streptomycin (Wisent). CEM.NKr CCR5+ cells (NIH AIDS reagent program) were maintained at 37 °C under 5% CO_2_ in Roswell Park Memorial Institute (RPMI) 1640 medium (GIBCO) containing 10% FBS and 100 μg/mL of penicillin-streptomycin. CEM.NKr CCR5+ cells stably expressing the SARS-CoV-2 S glycoproteins were previously reported [25].

### 2.5. Plasmids

The plasmids encoding the different SARS-CoV-2 S variants (D614G, Delta, BA.1 and BA.2) were previously described [15,16,27,32]. The pIRES2-EGFP expressing plasmid was purchased from Clontech, Mountain View, CA, USA (Cat# 6029-1).

### 2.6. Protein Expression and Purification

FreeStyle 293F cells (Invitrogen) were grown in FreeStyle 293F medium (Invitrogen) to a density of 1 × 10^6^ cells/mL at 37 °C with 8% CO_2_ with regular agitation (150 rpm). Cells were transfected with a plasmid coding for SARS-CoV-2 S RBD [32] using ExpiFectamine 293 transfection reagent, as directed by the manufacturer (Invitrogen). One week later, cells were pelleted and discarded. Supernatants were filtered using a 0.22 μm filter (Thermo Fisher Scientific, Waltham, MA, USA). The recombinant RBD proteins were purified by nickel affinity columns, as directed by the manufacturer (Invitrogen). The RBD preparations were dialyzed against phosphate-buffered saline (PBS) and stored in aliquots at −80 °C until further use. To assess purity, recombinant proteins were loaded on SDS-PAGE gels and stained with Coomassie Blue.

### 2.7. Enzyme-Linked Immunosorbent Assay (ELISA) and RBD Avidity Index

The SARS-CoV-2 RBD ELISA assay used was previously described [32,33]. Briefly, recombinant SARS-CoV-2 S RBD proteins (2.5 μg/mL), or bovine serum albumin (BSA) (2.5 μg/mL) as a negative control, were prepared in PBS and were adsorbed to plates (MaxiSorp Nunc) overnight at 4 °C. Coated wells were subsequently blocked with blocking buffer (Tris-buffered saline (TBS) containing 0.1% Tween20 and 2% BSA) for 1 h at room temperature. Wells were then washed four times with washing buffer (Tris-buffered saline (TBS) containing 0.1% Tween20). CR3022 mAb (50 ng/mL) or a 1/500 dilution of plasma were prepared in a diluted solution of blocking buffer (0.1% BSA) and incubated with the RBD-coated wells for 90 min at room temperature. Plates were washed four times with washing buffer followed by incubation with a secondary Abs (diluted in a diluted solution of blocking buffer (0.4% BSA)) for 1 h at room temperature, followed by four washes. To calculate the RBD-avidity index, we performed a stringent ELISA, where the plates were washed with a chaotropic agent, 8 M of urea, added to the washing buffer. HRP enzyme activity was determined after the addition of a 1:1 mix of Western Lightning oxidizing and luminol reagents (Perkin Elmer Life Sciences, Hopkinton, MA, USA). Light emission was measured with a LB942 TriStar luminometer (Berthold Technologies, Black Forrest, Germany). A signal obtained with BSA was subtracted for each plasma and was then normalized to the signal obtained with CR3022 present in each plate. The seropositivity threshold was established using the following formula: mean of pre-pandemic SARS-CoV-2 negative plasma + (3 standard deviation of the mean of pre-pandemic SARS-CoV-2 negative plasma).

### 2.8. Cell Surface Staining and Flow Cytometry Analysis

293T cells were co-transfected with a GFP expressor (pIRES2-GFP, Clontech, Mountain View, CA, USA) in combination with plasmids encoding the full-length S of SARS-CoV-2 variants; 48 h post transfection, S-expressing cells were stained with the CV3-25 Ab [28] or plasma (1/500 dilution). AlexaFluor-647-conjugated goat anti-human IgG Abs (1/1000 dilution) was used as a secondary Abs. The percentage of transfected cells (GFP+ cells) was determined by gating the living cell population based on viability dye staining (Aqua Vivid, Invitrogen). Samples were acquired on an LSRII cytometer (BD Biosciences, San Jose, CA, USA) and data analysis was performed using FlowJo v10.7.1 (Tree Star). The seropositivity threshold was established using the following formula: (mean of pre-pandemic SARS-CoV-2 negative plasma + (3 standard deviation of the mean of pre-pandemic SARS-CoV-2 negative plasma). The conformational-independent S2-targeting mAb CV3-25 was used to normalize S expression. CV3-25 was shown to effectively recognize all SARS-CoV-2 S variants [34].

### 2.9. ADCC Assay

The ADCC assay was previously described [25,35]. For evaluation of anti-SARS-CoV-2 antibody-dependent cellular cytotoxicity (ADCC), parental CEM.NKr CCR5+ cells were mixed at a 1:1 ratio with CEM.NKr cells stably expressing a GFP-tagged full-length SARS-CoV-2 S (CEM.NKr.SARS-CoV-2.S cells). These cells were stained for viability (AquaVivid; Thermo Fisher Scientific, Waltham, MA, USA) and cellular dyes (cell proliferation dye eFluor670; Thermo Fisher Scientific), to be used as target cells. Overnight rested PBMCs were stained with another cellular marker (cell proliferation dye eFluor450; Thermo Fisher Scientific) and used as effector cells. Stained target and effector cells were mixed at a ratio of 1:10 in 96-well V-bottom plates. Plasma (1/500 dilution) or monoclonal antibody CR3022 (1 μg/mL) were added to the appropriate wells. The plates were subsequently centrifuged for 1 min at 300× *g*, and incubated at 37 °C, 5% CO_2_ for 5 h before being fixed in a 2% PBS-formaldehyde solution. ADCC activity was calculated using the formula: [(% of GFP+ cells in Targets plus Effectors) − (% of GFP+ cells in Targets plus Effectors plus plasma/antibody)]/(% of GFP+ cells in Targets) × 100 by gating on transduced live target cells. All samples were acquired on an LSRII cytometer (BD Biosciences) and data analysis was performed using FlowJo v10.7.1 (Tree Star). The specificity threshold was established using the following formula: (mean of pre-pandemic SARS-CoV-2 negative plasma + (3 standard deviation of the mean of pre-pandemic SARS-CoV-2 negative plasma).

### 2.10. Microneutralization

Microneutralization was conducted as previously described [31]. Briefly, one day prior to infection, 2 × 10^4^ Vero E6 TMPRSS2 cells were seeded per well in the 96-well flat bottom plate and incubated overnight. Plasma dilutions ranged from 1/50 with 5-fold dilution until reaching 2 × 10^6^ and were performed in a separate 96-well culture plate using DMEM supplemented with penicillin (100 U/mL), streptomycin (100 μg/mL), HEPES, 0.12% sodium bicarbonate, 2% FBS and 0.24% BSA; 10^4^ × 50% tissue culture infectious dose (TCID50)/mL of SARS-CoV-2 D614G virus was prepared in DMEM + 2% FBS and combined with an equivalent volume of diluted plasma for one hour. After this incubation, all media were removed from the 96-well plate seeded with Vero E6 TMPRSS2 cells and virus: plasma mixture was added to each respective well at a volume corresponding to 600 TCID50 per well and incubated for one hour further at 37 °C. Both virus-only and media- only (MEM + 2% FBS) conditions were included in this assay. All virus–plasma supernatant was removed from the wells without disrupting the Vero E6 TMPRSS2 monolayer. Each diluted plasma (100 μL) was added to its respective Vero E6 TMPRSS2-seeded well in addition to an equivalent volume of MEM + 2% FBS and was then incubated for 48 h. Media were then discarded and replaced with 10% formaldehyde for 24 h to cross-link the Vero E6 TMPRSS2 monolayer. Formaldehyde was removed from the wells and subsequently washed with PBS. Cell monolayers were permeabilized for 15 min at room temperature with PBS + 0.1% Triton X-100, washed with PBS and then incubated for one hour at room temperature with PBS + 3% non-fat milk. An anti SARS-CoV-2 nucleocapsid protein (Clone 1C7, Bioss Antibodies) primary antibody solution was prepared at 1 μg/mL in PBS + 1% non-fat milk and added to all wells for one hour at room temperature. Following extensive washing (3×) with PBS, an anti-mouse IgG HRP secondary antibody (ThermoFisher) solution was formulated in PBS + 1% non-fat milk. One hour post room temperature incubation, wells were washed with 3× PBS, substrate (ECL) was added and an LB941 TriStar luminometer (Berthold Technologies) was used to measure the signal of each well.

## 3. Results

### 3.1. Elicitation of Anti-RBD-IgG and Their Associated Avidity against SARS-CoV-2 after mRNA Vaccination in PLWH

We first assessed the levels of anti-RBD IgG in PLWH stratified by their CD4 count (Table 1). We performed an ELISA assay previously described [16,25,29,33,36] with plasma collected shortly after mRNA SARS-CoV-2 vaccinal boost (4 weeks after the second dose) and later after the second mRNA SARS-CoV-2 vaccinal boost (12 weeks after the third dose) to evaluate the stability of the anti-RBD IgG over time (Figure 1A).

Four weeks after the second dose, we observed differences in the levels of anti-RBD antibodies in individuals with a CD4 count < 250/mm^3^ compared to individuals with a CD4 count between 250 and 500/mm^3^ (Figure 1B). In contrast, no differences were observed between the individuals with a CD4 count < 250/mm^3^ and individuals with a CD4 count > 500/mm^3^. Twelve weeks after the third dose, the levels of anti-RBD IgG were lower in individuals with a CD4 count < 250/mm^3^ compared to the other two groups (Figure 1B). We also observed that the levels of antibody between the two timepoints decreased for the group of donors with a CD4 count < 250/mm^3^, although not statistically significant, and donors with a CD4 count 250–500/mm^3^. In contrast, the antibody levels remained stable over time for the individuals with a CD4 count > 500/mm^3^ (Figure 1B).

While the levels of antibodies are important, their avidity is also an important parameter of the humoral response. Thus, we measured the avidity of vaccine-elicited antibodies by a previously described ELISA assay [16,37,38,39]. Briefly, plasma samples were tested by ELISA with or without stringent washes in parallel. The stringent washing buffer was supplemented with a chaotropic agent (8 M urea), allowing the antibodies with the highest anti-RBD affinity to remain bound to the RBD. Anti-RBD antibodies were detected with HRP-conjugated anti-human IgG. Four weeks after the second dose, we observed that the donors with a CD4 count < 250/mm^3^ had an RBD-specific IgG with lower avidity compared to the two other groups (Figure 1C). However, no statistical differences in the avidity of the anti-RBD IgG antibodies were observed between the three groups. Twelve weeks after the third dose, the avidity remained similar to what we observed 4 weeks after the second dose for the group with a CD4 count < 250/mm^3^. For the two other groups, we observed a decrease in avidity levels compared to the first timepoint, but this was not significant. No major differences in avidity were observed between the groups 12 weeks after the third dose (Figure 1C).

### 3.2. Capacity of Antibodies Elicited in PLWH to Bind Different SARS-CoV-2 Variants of Concern Spikes

We next assessed the capacity of vaccine-elicited antibodies to recognize the Spikes from different VOCs. We incubated plasma from donors with HEK 293T cells expressing D614G, Delta or Omicron subvariant Spikes. Spike expression levels were normalized to the signal obtained with the conformationally independent anti-S2 neutralizing CV3-25 antibody that efficiently recognized the four Spikes, as previously described [16,26,29,34,40]. The group with a CD4 count < 250/mm^3^ recognized less efficiently the D614G, Delta, BA.1 and BA.2 Spikes compared to the individuals with a CD4 count between 250 and 500/mm^3^ and individuals with a CD4 count > 500/mm^3^ 4 weeks after the second dose. However, we only observed statistically significant differences between the individuals with a CD4 count < 250/mm^3^ and individuals with a CD4 count between 250 and 500/mm^3^ (Figure 2A). Similar results were observed 12 weeks after the third dose, where individuals with a CD4 count < 250/mm^3^ had less anti-RBD antibodies able to recognize the different Spikes compared to the individuals with a CD4 count 250–500/mm^3^ and a CD4 count > 500/mm^3^, except for the BA.2 Spike where no statistical differences were observed between the three groups (Figure 2A). Spike recogniton by the antibodies elicited after the mRNA vaccination was stable between the two timepoints (Figure 2A). We further assessed the capacities of antibodies to recognize the different Spikes 4 weeks after the second dose or 12 weeks after the third dose (Figure 2B). No significant differences in the recognition of the different Spikes were observed at both timepoints for the group of individuals with a CD4 count < 250/mm^3^ (Figure 2B). The individuals with a CD4 count between 250 and 500/mm^3^ had antibodies that less recognized the BA.1 and BA.2 Spikes compared to D614G 4 weeks after the second dose and 12 weeks after the third dose (Figure 2B). In contrast, antibodies from donors with a CD4 count > 500/ mm^3^ were less efficient at recognizing the BA.2 Spike compared to D614G 4 weeks after the second dose, but showed differential recognition between all the Spikes except between BA.1 and BA.2 12 weeks after the third dose (Figure 2B).

### 3.3. Fc-Effector Functions and Neutralization against SARS-CoV-2 Spike of Vaccine-Elicited Antibodies

Lastly, we evaluated the ability of the antibodies elicited after mRNA vaccination to mediate Fc-effector function, such as ADCC and microneutralization of the live SARS-CoV-2 virus, as previously described [25,32,33,36,40,41]. At 4 weeks after the second dose, antibodies from individuals with a CD4 count < 250/mm^3^ depicted less ADCC response than antibodies from individuals with a CD4 count > 250/mm^3^ (Figure 3A). However, these differences were not observed 12 weeks after the third dose. Moreover, we observed no statistical differences in the capacity of these antibodies to mediate microneutralization of the live SARS-CoV-2 D614G virus 12 weeks after the third dose between the three groups (Figure 3B).

## 4. Discussion

In the general population, humoral responses are efficiently generated after the SARS-CoV-2 mRNA vaccine and increased after vaccinal boosts [16,29,42]. These responses are important since the neutralizing activity of antibodies elicited after vaccination is central to protect and predict the immune response to the SARS-CoV-2 infection [33,43,44]. We previously reported that PLWH having a CD4 count < 250/mm^3^ elicited less anti-RBD IgG than PLWH with a CD4 count > 250/mm^3^ after the first dose of the SARS-CoV-2 mRNA vaccine [24]. In this study, we measured the antibody levels after the second and third dose of the mRNA vaccine in the same cohort. We observed less anti-RBD IgG levels in individuals with a CD4 count < 250/mm^3^ 4 weeks after the second dose and 12 weeks after the third dose (Figure 1B). This was more pronounced in two individuals with a CD4 count < 200/mm^3^. However, despite the lower anti-RBD IgG levels, we observed no statistical differences in the avidity of the antibodies elicited 4 weeks after the second dose or 12 weeks after the third dose between the three different groups (Figure 1C). This suggests that B cell maturation and somatic hypermutation were unaffected by the CD4 count/mm^3^ of PLWH. This was not the case for the immunocompromised population [45].

With the increased rate of VOCs, several studies have shown decreased neutralizing capacities of antibodies elicited after mRNA vaccination against the different Spikes [15,42,46]. However, antibodies’ Fc-effector functions, such as ADCC, are part of the immune response and help clear the viral infection [25,40,47]. These functions were shown to be less affected than neutralization against VOCs in the general population, indicating the importance of Fc-effector functions in maintaining immune responses against the rising SARS-CoV-2 VOCs [48]. While we observed a lower ADCC response mediated by antibodies 4 weeks after the second dose between PLWH with a CD4 count < 250/mm^3^ and PLWH with a CD4 count between 250 and 500/mm^3^, this difference disappeared after the third dose, supporting recent findings [22,49,50]. Finally, we observed no differences between the capacity of these antibodies to neutralize live SARS-CoV-2 in the different groups after the third dose of the mRNA vaccine. This further supports the idea that despite lower elicitation of anti-RDB IgG after the third dose of the mRNA vaccine, the elicited antibodies are able to efficiently neutralize SARS-CoV-2, in agreement with what is observed in other immunocompromised individuals [51,52].

The small sample size for the group with low CD4 is a limitation of our study. It could be interesting to study the differential immune responses in individuals within the lower CD4 strata (below 200/mm^3^), however, in our population, people with a very low CD4 often have concurrent illnesses which would have affected immune responses and therefore, they could not be enrolled in our study.

## 5. Conclusions

In this study, we showed that PLWH having a CD4 count < 250/mm^3^ had lower levels of anti-RBD IgG and that these antibodies recognized less efficiently the different VOCs compared to PLWH having a CD4 count > 500/mm^3^. However, no major differences in the avidity, ADCC response and live virus neutralizing capacities of these antibodies were observed between the three groups 12 weeks after the third dose. Overall, our work demonstrates the importance of mRNA vaccine boosting in immunocompromised individuals presenting low levels of CD4.

## Figures and Tables

**Figure 1 viruses-15-02004-f001:**
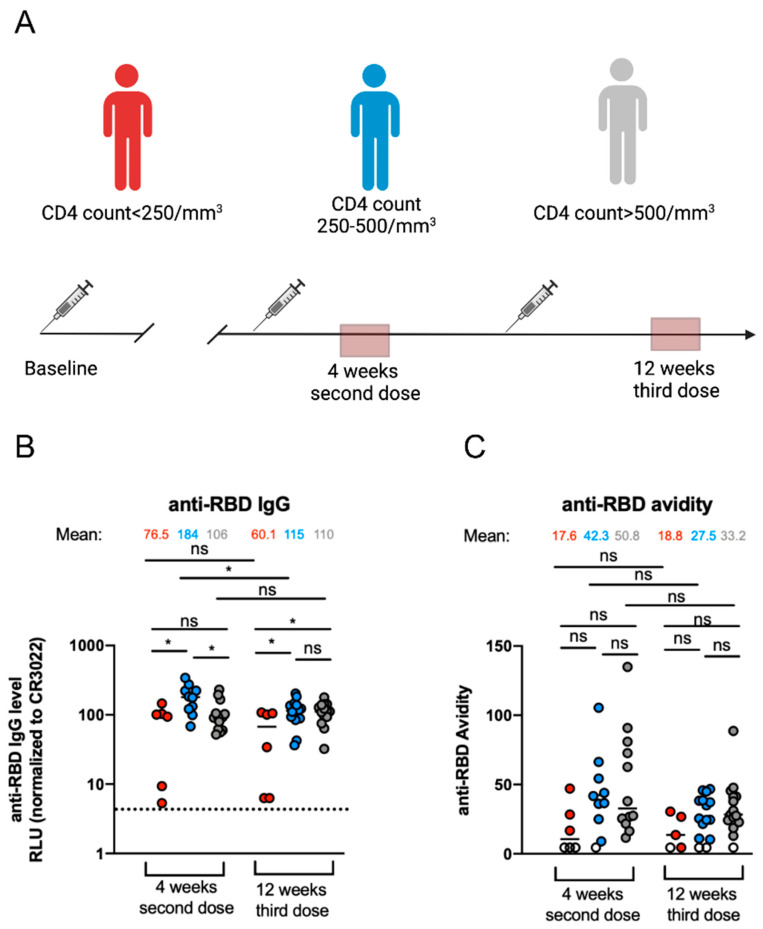
Anti-RBD IgG levels and associated anti-RBD avidity in PLWH 4 weeks after their second dose or 12 weeks after their third dose of mRNA SARS-CoV-2 vaccine. (**A**) SARS-CoV-2 vaccine cohort design. The red boxes represent the different timepoints. (**B**,**C**) Plasma samples from PLWH collected 4 weeks after the second dose or 12 weeks after the third dose of mRNA vaccine were incubated with recombinant SARS-CoV-2 RBD protein. Indirect ELISAs were performed with the plasma and RBD protein and anti-RBD Ab binding was detected using HRP-conjugated anti-human IgG. (**B**) RLU values obtained were normalized to the signal obtained with the anti-RBD CR3022 mAb present in each plate. (**C**) The anti-RBD avidity corresponded to the value obtained with the stringent (8 M urea) ELISA divided by that obtained without urea. PLWH with CD4 count < 250/mm^3^, CD4 count between 250 and 500/mm^3^ or CD4 count > 500/mm^3^ are represented by red, blue and gray points, respectively. Each circle identifies one donor. Values at threshold are represented by white circles. Dotlines represent seropositivity thresholds. Error bars indicate means. Statistical significance was tested using Mann–Whitney or unpaired *t* test. (* *p* < 0.05; ns, non-significant).

**Figure 2 viruses-15-02004-f002:**
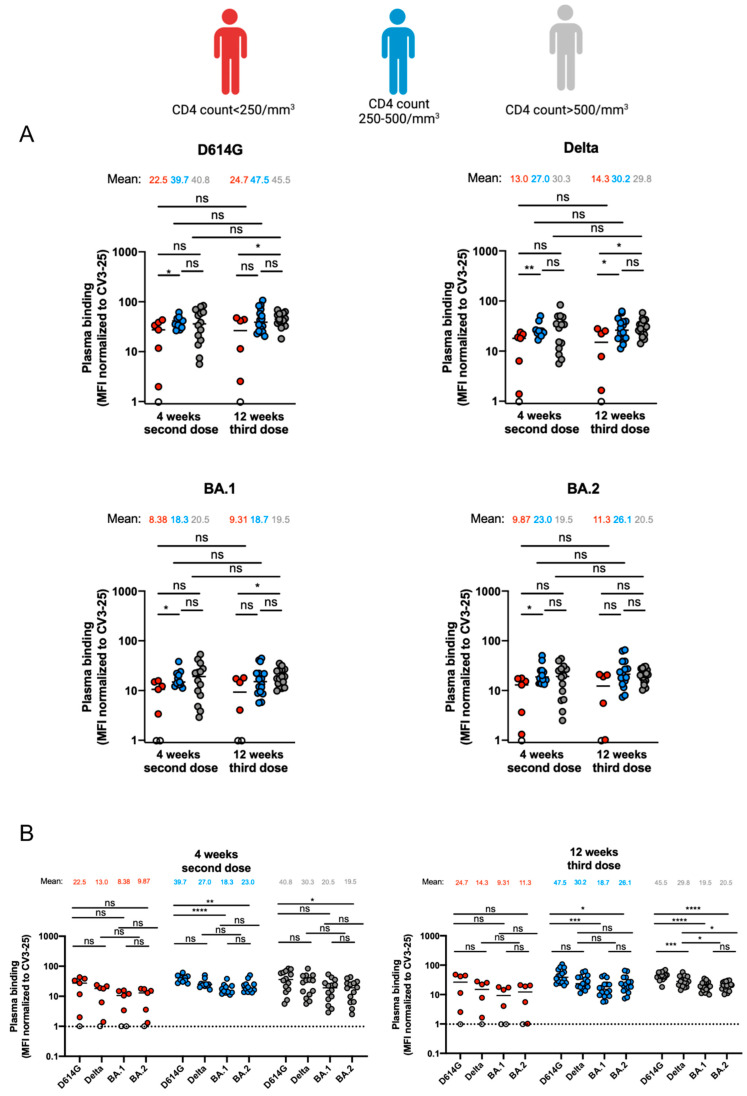
Binding of vaccine-elicited antibodies to SARS-CoV-2 Spike variants in PLWH. (**A**,**B**) HEK 293T cells were transfected with the indicated full-length Spikes from different SARS-CoV-2 variants (D614G, Delta, BA.1 and BA.2) and stained with the CV3-25 mAb or with plasma collected at 4 weeks after the second dose or 12 weeks after the third dose. Samples were analyzed by flow cytometry. The values represent the median fluorescence intensities (MFIs) normalized by CV3-25 mAb binding and presented as percentages of CV3-25 binding. PLWH with CD4 count < 250/mm^3^, CD4 count between 250 and 500/mm^3^ or CD4 count > 500/mm^3^ are represented by red, blue and gray points, respectively. Each circle identifies one donor. Values at threshold are represented by white circles. Seropositivity thresholds are plotted. Statistical significance was tested using (**A**) Mann–Whitney or unpaired *t* test or (**B**) Kruskal–Wallis or ordinary one-way ANOVA. Seropositivity thresholds are plotted. Error bars indicate means. * *p* < 0.05; ** *p* < 0.01; *** *p* < 0.001; **** *p* < 0.0001; ns, non-significant).

**Figure 3 viruses-15-02004-f003:**
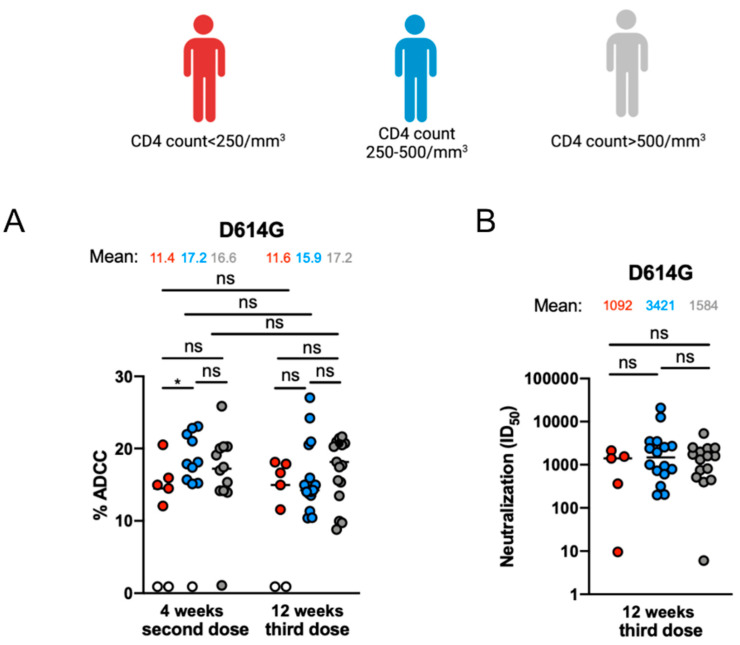
Fc-mediated effector functions and neutralization activities in PLWH. (**A**) In a FACS-based ADCC assay, CEM.NKr parental cells were mixed at a 1:1 ratio with CEM.NKr-Spike cells which were used as target cells. (**B**) Neutralizing activity was measured by incubating authenthic virus D614G, with serial dilutions of plasma for 1 h at 37 °C before infecting Vero E6 TMPRSS2. Neutralization half-maximal inhibitory serum dilution (ID_50_) values were determined using a normalized non-linear regression using GraphPad Prism software (Prism 10.0.1). PLWH with CD4 count < 250/mm^3^, CD4 count between 250 and 500/mm^3^ or CD4 count > 500/mm^3^ are represented by red, blue and gray points, respectively. Each circle identifies one donor. Values at threshold are represented by white circles. Seropositivity thresholds are plotted. Statistical significance was tested using Mann–Whitney or unpaired *t* test. (* *p* < 0.05; ns, non-significant).

**Table 1 viruses-15-02004-t001:** Characteristics of the PLWH cohort.

			CD4 Count/mm^3^		
		Entire Cohort	Below 250	250–500	Above 500
Number		40	7	16	17
Median Age		(25–77)	52 (25–77)	49 (40–62)	47 (35–60)
Sex	Male (n=)	33	5	14	14
	Female (n=)	7	2	2	3

## Data Availability

The data are contained within the article.

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
