# Peer review of "Humoral Responses Elicited by SARS-CoV-2 mRNA Vaccine in People Living with HIV"

_viruses, 2023, doi:10.3390/v15102004_

Round 1

Reviewer 1 Report

In this paper the authors evaluated the vaccine immunology elicited after 2 or 3 doses of mRNA Sars Cov 2 vaccine in PLWH stratified by their CD4 count. The authors found that after 2 doses of vaccine PLWH with CD4 count < 250 mm3 had lower level of anti RDB IgG antibodies compared with those with Cd4 count > 250mm3   The paper reaches conclusions that have already been partially demonstrated (vaccine 2022, Naut); AIDS 2023 Costiniuk) Moreover, the sample size in this study is not large

However, the work is well written .

Author Response

Dear Reviewer,

Thanks you for reviewing our manuscript and for your insightful comments.  We are pleased to resubmit a new version of the manuscript taking into consideration the reviews as indicated below.

Reviewer #1

In this paper the authors evaluated the vaccine immunology elicited after 2 or 3 doses of mRNA Sars Cov 2 vaccine in PLWH stratified by their CD4 count. The authors found that after 2 doses of vaccine PLWH with CD4 count < 250 mm3 had lower level of anti RDB IgG antibodies compared with those with Cd4 count > 250mm3   The paper reaches conclusions that have already been partially demonstrated (vaccine 2022, Naut); AIDS 2023 Costiniuk) Moreover, the sample size in this study is not large

However, the work is well written.

Reviewer 1 is right in the fact that this work is an extension of our previous study (Nault et al, Vaccine 2022), in which we had characterized the immune response of HIV-infected individuals after one dose of vaccine.  In this manuscript, we have characterized in depth the longitudinal immune responses of several doses of COVID-19 vaccine in the same cohort.

Thank you again for considering our manuscript, we look forward to your decision.

Reviewer 2 Report

Slightly used in the abstract is not a meaningful description and should be replaced by the use of statistical measures.

This is an important paper showing quite an appreciable immunogenicity of the mRNA vaccine in PLWH  indicating that in the PLWH examined a vaccine produced an immune response that is quite long-lasting, which we learned with some relief. In addition, the methodology used is appreciable and the research is well presented.

However, there are two points to be addressed: In 7 PLWH two had CD4+ cells below 250/cu mm all except two produced a similar level of RBD Abs as other  PLWH with higher count of CD4+ cells in the blood. Therefore, those two outliners should be additionally examined as to why they so poorly generated post-vaccination antibodies looking for the presence of abnormalities affecting abs response starting from the exact enumeration of CD4+ cells in the blood.  If they had CD4+ cells below 200/ul of the blood it would speak for the presence of a profound immune system deficiency situating them in the area associated with AIDS diagnosis. Therefore In the discussion section the rationale behind choosing the level of 250 CD4+ cells per ul in blood and why a lower limit of the number of CD4+ cells in the blood was not assigned in the below 250 CD4+ cells per should be discussed.

I believe that my suggestions should be considered before being accepted for publication.

This is an important paper showing quite an appreciable immunogenicity of the mRNA vaccine in PLWH, well written.

Author Response

Dear Reviewer,

Thanks you for reviewing our manuscript and for your insightful comments.  We are pleased to resubmit a new version of the manuscript taking into consideration the reviews as indicated below.

Reviewer #2

Slightly used in the abstract is not a meaningful description and should be replaced by the use of statistical measures.

As requested, we have changed the wording and added the p-value.

 This is an important paper showing quite an appreciable immunogenicity of the mRNA vaccine in PLWH  indicating that in the PLWH examined a vaccine produced an immune response that is quite long-lasting, which we learned with some relief. In addition, the methodology used is appreciable and the research is well presented.

 We thank the reviewer for his/her positive assessment of our work

 However, there are two points to be addressed: In 7 PLWH two had CD4+ cells below 250/cu mm all except two produced a similar level of RBD Abs as other PLWH with higher count of CD4+ cells in the blood. Therefore, those two outliners should be additionally examined as to why they so poorly generated post-vaccination antibodies looking for the presence of abnormalities affecting abs response starting from the exact enumeration of CD4+ cells in the blood.  If they had CD4+ cells below 200/ul of the blood it would speak for the presence of a profound immune system deficiency situating them in the area associated with AIDS diagnosis.

The reviewer is correct, these two individuals had low CD4 counts:  160 and 170 respectively.  We would like to highlight that in our cohort individuals with concurrent illnesses which could have affected immune responses were not be enrolled in our study.  This was clarified in the revised application.

Therefore In the discussion section the rationale behind choosing the level of 250 CD4+ cells per ul in blood and why a lower limit of the number of CD4+ cells in the blood was not assigned in the below 250 CD4+ cells per should be discussed.

I believe that my suggestions should be considered before being accepted for publication.

 In our cohort of treated HIV-infected individuals, very few have low CD4 counts.  We have chosen the 250 cut-off in order to have a sufficient number of patients to analyse who did not have concurrent illnesses.  We have added a sentence in the discussion explaining that choice.

Thank you again for considering our manuscript, we look forward to your decision.

Reviewer 3 Report

In this paper, the authors conducted a longitudinal study of vaccine immunogenicity elicited after two and three doses of mRNA SARS-CoV-2 vaccine in PLWH stratified by their CD4 count. They also assessed the quantity and quality of anti-SARS-CoV-2 antibodies elicited after the second and third dose of vaccination. 

The present paper can be accepted for the publication after some major revisions.

1. The introduction section is short. it can be improved by adding the         motivation and novelty of your study.

2. Some accronyms should be defined.

3. Some figures are not clear.

4. Compare your results with others existing in the literature.

5. Check the language and punctuation of the manuscript.

6. Check and unify the citation of references.

Check the language and punctuation of the manuscript.

Author Response

Dear Reviewer,

Thanks you for reviewing our manuscript and for your insightful comments.  We are pleased to resubmit a new version of the manuscript taking into consideration the reviews as indicated below.

Reviewer #3

In this paper, the authors conducted a longitudinal study of vaccine immunogenicity elicited after two and three doses of mRNA SARS-CoV-2 vaccine in PLWH stratified by their CD4 count. They also assessed the quantity and quality of anti-SARS-CoV-2 antibodies elicited after the second and third dose of vaccination.

The present paper can be accepted for the publication after some major revisions.

  1. The introduction section is short. it can be improved by adding the motivation and novelty of your study.

We have added a sentence to this effect.

  1. Some accronyms should be defined.

We have defined all acronyms. 

  1. Some figures are not clear.

Reviewers #1 and #2 didn’t object to the figures.  Since reviewer #3 didn’t indicate which figure and what should be changed, we prefer to keep the figures as they are. 

  1. Compare your results with others existing in the literature.

In the discussion section, we compare our results to other published studies. 

  1. Check the language and punctuation of the manuscript.

The language and punctuation has been checked.

  1. Check and unify the citation of references.

The references are cited as per guidelines.  

Thank you again for considering our manuscript, we look forward to your decision.

Round 2

Reviewer 1 Report

the comments on the paper were already forwarded a couple of weeks ago, in which it was reported that the work did not add important news on the subject eg see HIV Med 2023 ; furthermore the caseload was rather limited. Nonetheless the paper is well written

Un increased number of cases will be appropriated

it is ok 

Reviewer 3 Report

The paper can be accepted.

The paper can be accepted.
